# Farmer perceptions and willingness to pay for novel livestock pest control technologies: A case of tsetse repellent collar in Kwale County in Kenya

Beatrice W. Muriithi[1]*, Nancy G. Gathogo[2], Gracious M. Diiro[1], Michael M. Kidoido[1], Michael Nyangánga Okal[1], Daniel K. Masiga[1]

1 International Centre of Insect Physiology and Ecology (*icipe*), Nairobi, Kenya, 2 The Salvation Army, Nairobi, Kenya

* bmuriithi@icipe.org

**Data Availability Statement:** The authors confirm that all data are fully available without restriction.

## Abstract

Tsetse-transmitted Animal African Trypanosomosis (AAT) is one of the most important constraints to livestock development in Africa. Use of trypanocides has been the most widespread approach for the management of AAT, despite the associated drug resistance and health concerns associated with drug metabolites in animal products. Alternative control measures that target tsetse fly vectors of AAT, though effective, have been hard to sustain in part because these are public goods applied area-wide. The International Centre of Insect Physiology and Ecology (*icipe)* and partners have developed and implemented a novel tsetse repellent collar (TRC) applied on animals to limit contact of tsetse flies and livestock, thereby reducing AAT transmission. The TRC has now advanced to commercialization. A household-level survey involving 632 cattle keeping households, was conducted in Shimba Hills region of Kwale County, where field trials of the TRC have been previously conducted to assess farmers' knowledge, perception, and practices towards the management of tsetse flies, their willingness to pay (WTP) for the TRC, and factors affecting the WTP. Almost all the respondents (90%) reported that tsetse flies were the leading cattle infesting pests in the area. About 22% of these correctly identified at least four AAT clinical signs, and even though many (68%) used trypanocidal drugs to manage the disease, 50% did not perceive the drug as being effective in AAT management (50%). Few respondents (8%) were aware of the harmful effects of trypanocidal drugs. About 89% of the respondents were aware of *icipe* TRC, and 30% of them were using the field trial collars during the survey. Sixty-three (63%) of them were willing to pay for the TRC at the same cost they spend treating an animal for AAT. On average farmers were willing to pay KES 3,352 per animal per year. Male educated household heads are likely to pay more for the TRC. Moreover, perceived high AAT prevalence and severity further increases the WTP. Wider dissemination and commercialization of the herd-level tsetse control approach (TRC) should be encouraged to impede AAT transmission and thus enhance food security and farm incomes among the affected rural communities. Besides the uptake of TRC can be enhanced through training, especially among women farmers.

All relevant data are within the manuscript and its Supporting Information files.

**Funding:** The authors would like to acknowledge financial support from the European Union (Grant number DCI-FOOD/2014/346-739 to DKM) through The Integrated Biological Applied Research Programme (IBCARP), Tsetse Collar component; and the Biovision (BV) foundation (Grant number (BV HH-07/2016-18 to DKM). We also acknowledge the icipe core funding from UK's Foreign, Commonwealth & Development Office (FCDO); the Swedish International Development Cooperation Agency (Sida); the Swiss Agency for Development and Cooperation (SDC); the Federal Democratic Republic of Ethiopia; and the Government of the Republic of Kenya. The funders had no role in study design, data collection and analysis, decision to publish, or preparation of the manuscript.

**Competing interests:** The authors have declared that no competing interests exist.

## Author summary

Animal African Trypanosomosis is a tropical disease that is of economic importance in Sub-Saharan Africa. The livestock sub-sector supports approximately 600 million small-holders in developing countries through employment, income from livestock products, and improving crop productivity through draft power and manure. Efforts to reduce rural poverty and improve food security and nutrition, therefore, require utilizing livestock to their full potential. Trypanosomosis results in anemia, emaciation, productivity loss, and mortality, and remains a leading constraint to livestock development in Africa. To reduce the risks associated with the use of trypanocides, alternative control measures that target the vector- tsetse fly need to be developed and widely disseminated. The International Centre of Insect Physiology and Ecology (*icipe*) and partners have—developed and implemented a novel tsetse repellent collar that shields animals from getting into contact with the vector, thereby preventing trypanosomosis transmission. The collar has now advanced to a commercialization stage. We conducted community and household-level surveys to assess farmer's knowledge, perception, and practices regarding tsetse pest and trypanosomosis, and their willingness to pay for the novel tsetse repellent collar. We found that the pest is the main cattle production constraint and the cause of the associated disease, although there exists a gap in the identification of the clinical signs of the disease. Besides, most farmers rely on trypanocidal drugs for the treatment of their animals despite their human health and environmental risks. However, the majority were willing to buy the newly developed canvas collar. A male head of a household with a higher level of education is likely to pay more for the novel tsetse repellent collar. Besides, perception of high prevalence and severity of AAT is likely to increase the willingness to pay for the herd-level technology. The findings emphasize the need for wider dissemination and commercialization of the tsetse repellent collar technology to reduce trypanosomosis transmission and hence enhance food security and farm incomes in the affected regions in Africa.

## 1 Introduction

Livestock is a major sector in many developing countries, contributing up to 30% of their agricultural GDP [1]. The sector provides food and income to both the rural and urban population and is a source of manure and draft power used in crop production. Moreover, livestock is a source of cultural heritage and way of life [2, 3]. In Kenya, the livestock sub-sector contributes about 12% of the total agricultural GDP [4] and 14% of the agricultural labour force [5]. The northern Arid and Semi-Arid areas of Kenya dominate livestock production in the country, accounting for over 60% of the national beef cattle population [6]. Productivity of the livestock subsector in Kenya is, however, very poor with most of the livestock weighing less than the minimum market weight of 350Kg [6]. Whereas factors that constrain livestock productivity are many (including drought, lack of fodder, pests, and diseases), the Animal African trypanosomosis (AAT) transmitted by tsetse is considered to be the most important in Africa [7, 8].

ATT is a disease endemic in the SSA [7, 8] and is estimated to cause three million cattle deaths in the region annually and more than 46 million cattle, risk contracting the disease [9]. Direct production losses associated with the disease amount to $1.2 billion each year [10]. Tsetse flies that transmit AAT occur in landscapes of 37 SSA countries posing a major threat to the livelihoods of many households in these countries [11].

Whereas, scientists and development partners have devoted a lot of resources to control or eradicate the vector and the disease, cost-effective eco-friendly interventions to control AAT are scarce in the SSA [12]. Most of the disease control efforts have focussed on therapy to sick animals with trypanocidal drugs [13]. There is evidence of increasing trypanocides resistance and health and environmental risks related to drug toxicity and improper disposal of drug left-overs [14, 15]. Worse still, the development and use of vaccines against trypanosomosis remain futile [10, 14]. Eco-friendly vector control approaches that are available such as insecticide-impregnated targets [16, 17], tsetse fly traps, and pour-on technologies [18] are—expensive to implement and thus not widely adopted by the resource-poor livestock farmers in the SSA [19]. The International Centre of Insect Physiology and Ecology (*icipe)* and partners have over the last decade invested resources to develop novel eco-friendly vector control tools implemented both at community (area-wide strategies) and household (herd) level to increase efficiency, effectiveness, and safety in the reduction of AAT transmission [16]. The most notable one is tsetse repellents delivered through a wearable collar that cattle carry around. Efficacy studies show that the repellent collars are safe on livestock, humans, and the environment, and can effectively reduce AAT infections in animals by preventing tsetse bites [16].

Although the repellent collars technology (TRC) for AAT control exhibits potential economic and environmental benefits to the livestock farmers and the subsector in the SSA region [16], wide-scale commercialization and adoption of the technology will depend on farmers' pre-conceived perceptions, preferences, and their acceptance for the new technology. The objective of this study was to assess cattle keepers' perceptions of the AAT and its control and their willingness to pay for TRC for AAT control. The study used the contingent valuation (CV) method to elicit farmer's WTP for the TRC using data collected from cattle keepers in Kenya using Kwale county as a case study. This is the first study to empirically estimate the WTP for the TRC. This paper complements this previous literature by estimating the WTP for TRC, a relatively new technology whose potential demand has not been empirically estimated before. Moreover, this study assesses farmers' knowledge, perceptions, and attitudes towards the new technology. The concepts of knowledge, attitudes, and perceptions are widely applied in analyzing smallholder pest management decisions in developing countries (e.g. Lagerkvist et al., [20]; Schreinemachers et al., [21]). It is assumed that changes in farmers' practices regarding new technologies are the cumulative result of changes in farmers' knowledge, attitude, and perceptions [21, 22].

Our results showed that tsetse fly and the associated disease (AAT), remain an important economic constraint to cattle production in the study region. Besides, we found a knowledge gap in the identification of the clinical signs and symptoms associated with AAT, and that majority of the farmers rely on trypanocides drugs for the treatment of their animals. However, we found a positive willingness to purchase the newly developed canvas collar. With respect to farm and farmer characteristics, being male head of a household, and perceiving AAT prevalence and severity to be high, increased the willingness to pay for the repellent collar. These findings have important implications for wider dissemination and commercialization of the tsetse repellent collar technology. Training farmers, especially women, may enhance the uptake of TRC in tsetse prone areas in Kenya and beyond.

## 2 Materials and methods

### 2.1 Ethics statement

The study received ethical clearance from the Research Ethics Review Committee of *icipe*. Oral consent was sought from the respondents who were provided with sufficient information

about the research to allow them to make informed and independent decisions on their participation in the survey.

## 2.2 Conceptual framework

We conceptualize farmer WTP for TRC using the random utility framework in which producers are assumed to adopt a new agricultural innovation if the utility obtained is higher than what is currently in use, subject to factors of production [23]. We assess WTP using the Contingent Valuation (CV) approach, one of the survey-based stated preference methods used to elicit producers' evaluation of new non-market traded technologies. The CV method uses hypothetical survey questions to elicit respondents' WTP or accept public goods and services that are not traded in the marketplace [24–26]. The method generates useful information using a probability sample through either face-to-face or telephone interviews, and presents information about a product/innovation before asking respondents to state the amount they would pay to obtain product [25, 26]; the respondents consider their actual budget constraint when considering their WTP [25, 26]. The CV approach has been widely applied to study the demand for new agricultural technologies and innovations (e.g. Krishna and Qaim, [27]).

Elicitation of the WTP using survey-based CV method utilizes either single-bounded, double-bonded, and multi-bounded contingent valuation method [28]. In the single-bounded approach, the respondents are faced with a single question or a single bid value to which they accept or reject depending on their maximum WTP amount [29]. Alternatively, they can be assessed on the likelihood of paying for the product without attaching any price to it [30]. This method however requires large samples and may not result in efficient estimates [28]. This paper uses the double-bonded CV model, in which survey respondents are faced with two-sequence-bid offers. Unlike the single-bounded model, a second bid is presented to the respondent; a higher bid if the response was yes to the first bid, and a lower bid otherwise. The questions progressively narrow down the WTP, therefore providing more information about the respondents' WTP and leading to efficient WTP estimates [28, 31]. Multiple-bounded models offer multiple bids [32]. The approach is particularly useful when information about the potential bids is limited before the survey, therefore offering the possibility of including several options for uncertainty. However, it might be affected by bias that may occur at design or deciding on the range of bids to be included [33].

In the double-bonded model, in the first offer, respondents are asked whether they will accept or reject the bid value. In their second bid, respondents are then offered based on their first bid responses. If the respondent answered "yes" to the first bid ($B_i$), then he/she is presented with a higher bid amount in the second bid ($B_i^U$). However, if they answered "no" to the first bid, then a lower amount ($B_i^L$) is offered. The bidding process results in four possible responses: (1) both answers are "yes", denoted by $\pi^{yy}$, (2) a "yes" followed by a "no", denoted by $\pi^{yn}$, (3) a "no" answer followed by a "yes", denoted by $\pi^{ny}$, and (4) both answers are "no" ($\pi^{nn}$). The probability that both answers are "yes" indicates that the respondent's maximum WTP is higher than the highest bid offered ($\pi^{yy} (B_i, B_i^U) = Pr(B_i^U < maxWTP_i)$). The probability that the answers are "yes" to the first bid and "no" to the second bid indicates that the respondent's maximum WTP is higher than the first bid but lower than the second (higher) bid offered ($\pi^{yn} (B_i, B_i^U) = Pr (B_i < maxWTP_i < B_i^U)$). The probability that the answers are "no" to the first bid and "yes" to the second bid indicates that the respondent's maximum WTP is lower than the first bid but higher than the second (lower) bid offered ($\pi^{ny} (B_i, B_i^L) = Pr (B_i^L < maxWTP_i < B_i)$). The probability that both answers are "no" indicates that the respondent's WTP is lower than the second (lower) bid offered ($\pi^{yy} (B_i, B_i^L) = Pr(B_i^L < maxWTP_i)$) [28, 31]. If we consider N farmers were involved in the survey, and the $Bi$ bids

offered to the $i^{th}$ farmer, the log-likelihood function for the above set of responses can be expressed as follows:

$$ln\,L^D|(\theta) = \sum_{i=1}^{N}\{d_i^{yy}ln\pi^{yy}(B_i, B_i^u) + d_i^{yn}ln\pi^{yn}(B_i, B_i^u) + d_i^{ny}ln\pi^{ny}(B_i, B_i^l) + d_i^{nn}ln\pi^{nn}(B_i, B_i^l)(1)$$

where $d_i^{yy}$, $d_i^{yn}$, $d_i^{ny}$ and $d_i^{nn}$ are binary variables with the yes/no response to the first and second bid offers and $\pi$ represents the response probabilities for each of the four possible responses. The maximum likelihood function is estimated using the interval regression in STATA version 16.0. The regression results are used to estimate the mean WTP using robust bootstrapped standard errors [28], and the factors that influence WTP for tsetse repellent collars among cattle keepers are also identified.

## 2.3 Data and empirical strategy

**2.3.1 Study area, sampling procedure, and data.**    The study was conducted in Kwale County in the coastal region of Kenya. A purposive sampling method was used to select the sub-county, wards, and villages for the survey, based on the ongoing *icipe*'s field promotional campaign for tsetse control and AAT management. The county is one of the most tsetse-infested and AAT-affected areas in Kenya, with a prevalence rate of up to 56 percent in hotspots [11, 34]. It occupies an area of 8,267Km$^2$ with an estimated population of 866,820 people [35]. Mixed farming (crops and livestock) is the main economic activity of the area. Matuga Sub-county and subsequently three wards, namely: Kubo South, Tsimba/Golini, and Mkongani wards, were purposively selected. Twelve (12) villages were then selected from the three wards: eight from Kubo South, two from Mkongani, and one from Tsimba Golini (Fig 1). All

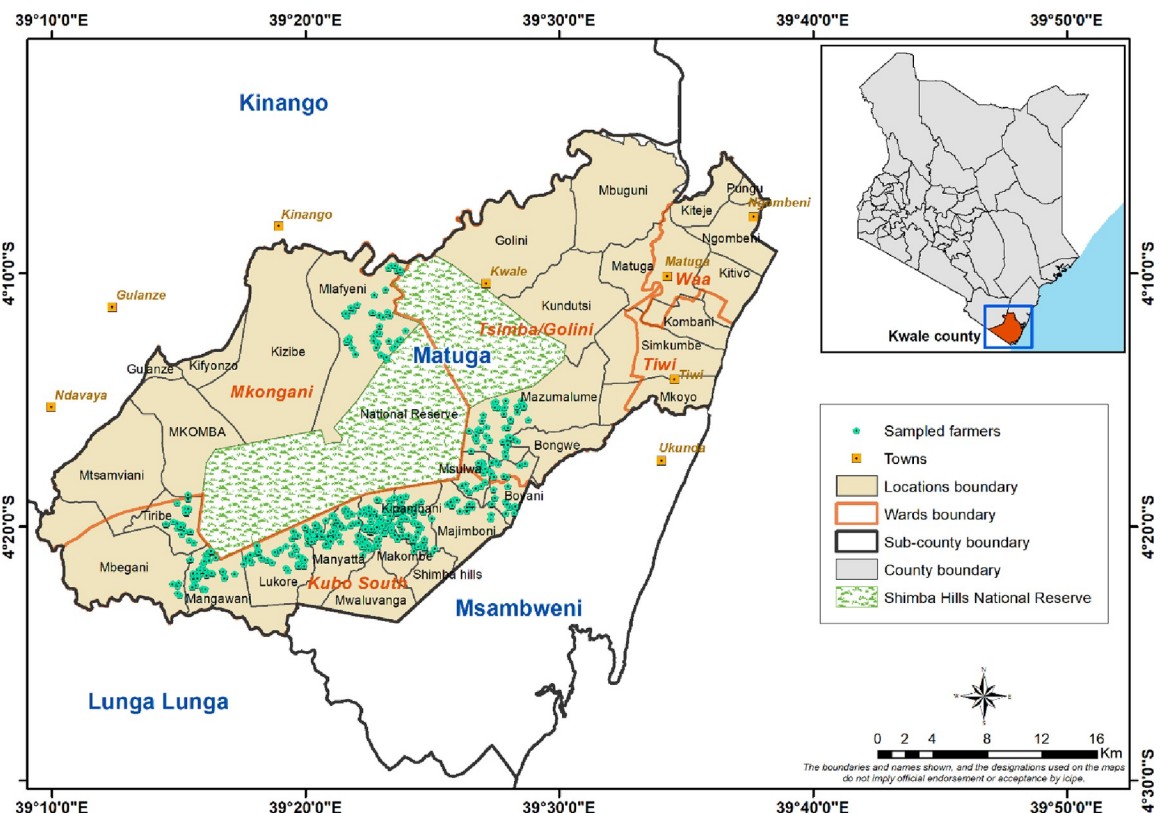

**Fig 1. Cartography: Emily Kimathi, GIS unit, icipe; The map was developed using the QGIS 3.16 software, https://qgis.org/en/site/forusers/download.html.**

the villages are adjacent to the Shimba Hills National Reserve (SHNR), reservoir of many tsetse flies and AAT alternative hosts, that perpetuate a high burden of tsetse and prevalence of AAT.

A list of cattle farmers from the 12 villages was obtained with the support of the front-line extension officers from the area. The list provided a sampling frame from which 632 households were randomly selected for interviews. The probability proportional to size (PPS) approach to determine the number of farmers to interview from each block.

Data were collected at the household level by well-trained enumerators using a pre-tested structured digital questionnaire programmed in CSPro 7.0 data collection software uploaded on CAPI-enabled devices. The questionnaire captured the following information: livestock production and marketing and related constraints, knowledge, perception, and management of tsetse and AAT including *icipe* TRC and WTP for the collar, socioeconomic and demographic characteristics, availability, and access to agricultural support services, markets, and market information, social capital and networks, and other contextual data. The survey was administered to the head of the household, and in their absence, the spouse, to ensure that the information provided was as accurate as possible. Besides, the enumerators had to understand and speak the local language to ensure they correctly communicated the questions to the selected household participants. The survey was conducted between September and October 2018. The study tool and protocol were reviewed and approved by the *icipe*'s research ethics review committee before the commencement of the survey. Respondent's consent was sought by the enumerators at the introduction of the survey to ensure voluntary and ethical participation by the farmers. Before the household level survey was conducted, qualitative information was obtained from the study area through focus group discussions (FDGs) and Key Informant Interviews (KIs). The qualitative survey complemented the design of the household-level survey and further enriched the analysis of WTP.

To capture the respondents' knowledge and perceptions of tsetse flies and AAT, including awareness and use of *icipe* TRC, respondents were first asked about their knowledge of the pest and the awareness of the various symptoms associated with AAT. Responses were then cross verified with a stack of photographs of various cattle infesting pests and a list of known clinical signs of AAT, as suggested by various experts from *icipe* and literature [36]. Without correcting them if they did not correctly identify a pest, respondents were then asked to explain how the pest affected cattle (symptoms) and how prevalent they considered them. A three-point rating scale (high, medium, and low) was used for this assessment. Once they had listed all the pests, and diseases and rated their severity and prevalence, respondents then ranked the pests and diseases based on their economic importance. To assess farmers' practices towards tsetse and AAT, farmers were asked to list various current methods used to control the pest and disease. Similarly, coloured photos of tsetse control methods, including tsetse fly traps, targets, and TRC were used to establish their usage in the villages.

**2.3.2 Elicitation of willingness to pay (WTP).** To elicit the willingness to pay for TRC, first, the farmers' awareness of the collar was assessed by asking whether they had ever heard of the technology. Irrespective of their awareness of status, respondents were presented with a short explanation of the scientific background of the collar technology. The narrative began by outlining the benefits of using the collars as an alternative to the commonly used trypanocides (see S1 Text). Photographs of the old and recently developed TRC (see photos (a) and (b) in S1 Fig), which is due for commercialization, were then presented to the farmers to enable them to envision the technology. One collar (and repellent sachet) would cost KES 350 ($3.5) and to be refilled every 6 weeks at approximately KES 125 ($1.25) (annual cost approximates KES 1,475) for one animal. After presenting the above narrative, the farmer was then asked to state how much it would cost to treat one animal per year for tsetse and AAT (without using any vector

control measures). This amount was the first bid of the contingent valuation, followed either by a higher or lower amount as explained earlier in the conceptual framework and S1 Text.

**2.3.3 Empirical model for determinants of WTP.** Based on the household utility maximization theory, we assess the explanatory factors that influence WTP for TRC. Generally, the probability of a farmer $i$ acquiring a TRC at a certain price $B_i$ can be expressed as a function of explanatory as follows.

$$\pi^y(B_i, \boldsymbol{x}_i) = \pi^y(v) \tag{2}$$

where $v$ is the index function with the preterminal relationship between the bid-offer $B_i$ and explanatory variables $\boldsymbol{x}_i$, which are assumed to be linear, such that:

$$v = \alpha - \rho B_i + \gamma' x_i + \boldsymbol{\varepsilon}_i \tag{3}$$

where $\boldsymbol{x_i}$ is the vector of explanatory variables used to explain WTP and include proxies for socioeconomic and demographic characteristics, availability and access to agricultural information and farmer support services, social capital and networks, and knowledge and perceptions on tsetse and AAT management. $\gamma'$ β is a vector of unknown parameters to be estimated, and $\boldsymbol{\varepsilon}_i$ is a random error term with mean zero and variance $D$. The parameters were estimated by maximizing the log-likelihood function of the outcomes in the bidding process. The maximum likelihood estimation generates $\gamma'$ and Sigma (**σ**) that are then used to derive WTP. Accordingly, the mean WTP is obtained as $E(WTP) = \hat{y}'\bar{x}$ [37, 38]. The selection of explanatory variables ($\boldsymbol{x_i}$) used in this study was derived from the theoretical and empirical literature on the adoption of agricultural innovations and the study context. The variables are broadly classified into five categories: *household characteristics* (including gender, age and education level of the household head, and family size) [39–41]; *household resources* (including livestock, farm size, the main occupation of the household head, annual income, and access to off-farm income, are important determinants of adoption of new technologies) [42, 43]; *access to market and institutional services* (training, government extension, and advisory services, credit, and market) [44, 45]; *social capital and networks (*membership in a rural institution); and *knowledge and perception indicators (*including awareness of the AAT clinical symptoms and perceived prevalence, negative effects of trypanocides and perceptions towards their effectiveness, and TRC awareness) [26, 27, 36].

# 3 Results

## 3.1 Socioeconomic characteristics of the sample of cattle keepers

Table 1 provides the definition and summary of the selected demographic, resource, and social capital and network characteristics hypothesized to affect the WTP for TRC. About 84% of the surveyed households were headed by males with a mean age of 53 years and average formal education of 7.4 years. On average, the surveyed households had six members. In agricultural technology adoption literature, female-headed households are often viewed to have limited access to productive resources in comparison with men [40]. We, therefore, hypothesize that male-headed households have higher WTP for TRC owing to their resource endowment. On the other hand, older household heads may become risk-averse as the time horizon in which to reap the benefits of adopting a new technology decrease and thus lower WTP for TRC. Better educated heads of households are viewed to possess greater human capital and technical skills that are often associated with the early adoption of technologies [39]. Therefore we expect them to have higher WTP. Large household size may indicate labour availability for farm activities that may facilitate new technology and thus higher WTP for TRC. For instance, the use of TRC enables farmers to graze their animals in tsetse-infested areas near the national

**Table 1. Descriptive summary of socio-economic characteristics of the sampled cattle keepers in Kwale County.**

| | | Sample n = 632 | | [95% Conf. interval] | |
|---|---|---|---|---|---|
| | | **Mean** | **Std. Dev.** | | |
| ***Household characteristics*** | | | | | |
| Gender | Sex of the household head (0 = female;1 = male) | 0.84 | 0.37 | 0.81 | 0.87 |
| Age | Age of household head (years) | 52.89 | 14.10 | 51.79 | 53.99 |
| Education level | Education of the household head (years) | 7.36 | 4.04 | 7.04 | 7.67 |
| Household size | Adult equivalents in the house (adult equivalent) | 2.90 | 1.00 | 2.82 | 2.98 |
| ***Household resources*** | | | | | |
| Livestock (TLU) | Small (< = 3.16) | 2.41 | 0.49 | 2.37 | 2.45 |
| | Medium (3.17–7.95) | 4.88 | 1.07 | 4.80 | 4.97 |
| | High (>6.95) | 10.85 | 4.64 | 10.48 | 11.21 |
| | *Mean TLU* | 5.76 | 3.96 | 5.45 | 6.07 |
| Farm Size (Hectare) | Small (<1.618778) | 1.13 | 0.41 | 1.10 | 1.16 |
| | Medium (1.619–4.047) | 2.75 | 0.72 | 2.69 | 2.81 |
| | High (>4.047) | 8.18 | 3.36 | 7.92 | 8.45 |
| | *Mean farm size* | 3.45 | 3.17 | 3.20 | 3.70 |
| Main occupation | Farming (%) | 0.78 | 0.41 | 0.75 | 0.81 |
| | Salaried (%) | 0.10 | 0.30 | 0.08 | 0.12 |
| | Self-employed (%) | 0.06 | 0.24 | 0.04 | 0.08 |
| | Casual (%) | 0.04 | 0.21 | 0.03 | 0.06 |
| Off-farm income | Access to income from all other sources except the farm (0 = No; 1 = Yes) | 0.67 | 0.47 | 0.63 | 0.70 |
| ***Access to market and institutional services*** | | | | | |
| Livestock training | Access to livestock management training within 12 months prior to the survey (0 = No; 1 = Yes) | 0.21 | 0.41 | 0.18 | 0.24 |
| Extension proximity | Distance to the nearest government veterinary extension office from residence (walking minutes) | 111.03 | 83.46 | 104.51 | 117.55 |
| Credit | Credit constrained (0 = No; 1 = Yes) | 0.46 | 0.50 | 0.42 | 0.50 |
| Market distance | Distance to the main farm produce (livestock and crops) market from residence (walking minutes) | 215.23 | 127.88 | 205.24 | 225.22 |
| ***Social capital and networks*** | | | | | |
| Rural institutions | Participate in rural institutions e.g. Producers Organization (0 = No;1 = Yes) | 0.87 | 0.34 | 0.84 | 0.89 |
| ***Knowledge and perceptions*** | | | | | |
| AAT clinical symptoms | Identify AAT symptoms correctly (count out of 4 major symptoms) | 0.22 | 0.41 | 0.19 | 0.25 |
| Negative chemical effects | Being aware of the negative effects of trypanocides (0 = No; 1 = Yes) | 0.08 | 0.28 | 0.06 | 0.11 |
| Trypanocides effectiveness | Perceived effectiveness of Trypanocides in AAT management (0 = No; 1 = Yes) | 0.50 | 0.50 | 0.46 | 0.54 |
| AAT prevalence | Perceived trypanosomiases prevalence (0 = Low 1 = High) | 0.16 | 0.37 | 0.13 | 0.19 |
| Aware of tsetse collar | Aware or ever used *icipe* (trial) tsetse repellent collar (0 = No; 1 = Yes) | 0.91 | 0.28 | 0.89 | 0.93 |

Source: Household survey

park, especially during the dry periods. This additional effort may require extra labour that could be made available by a larger household.

With respect to household resources, the surveyed households owned on average 5.8 tropical livestock units (TLU). Big herd size and full-time operation on the farm are expected to be an incentive to adopt the TRC to increase livestock efficiency. On average, each household owned about 3.45 hectares of land. Extant literature confirms the positive correlation between farm size and adoption of agricultural technologies, especially those that are crop productivity-improving [46]. Households with bigger land sizes might therefore be more willing to invest in TRC. Mixed crop farming was the main occupation for most household heads as reported by 78% of the survey respondent. This finding corroborates with Mbahin et al. [34], who noted that cattle in this area were mainly kept for draft power subsistence use to cultivate food and

cash crops and hire neighbours. About 67% of the households in our sample had access to off-farm income. While off-farm income may provide the required capital to finance innovation investment [43], it may also divert farmer's time and effort dedicated to agricultural activities, thus reducing investment in new agricultural technologies [42].

Access to market and institutional services are important in determining the adoption of farm productivity-enhancing technologies such as those for livestock pest management [44, 45], and thus likely to influence their WTP for TRC. About 21% of the surveyed households had ever participated in livestock training. Livestock management training is hypothesized to positively influence WTP as it provides information on the use and benefits associated with the technology [47]. Remoteness from extension services was noted among the sampled households who reported living nearly 2 hours of walking distance to the nearest government veterinary extension office from residence. Government extension services are important for enhancing access to knowledge about new agricultural technologies [39]. Therefore, longer distances would negatively influence knowledge of the innovation and thus reduce the WTP for TRC. Similarly, our sampled households seem to be located away from the nearest main farm produce (about 3.5 hrs walking distance). Limited market access is expected to negatively influence the farmer's WTP for TRC as it affects timely access to inputs and output disposal. About 46% of the respondents reported being credit-constrained. Households that are credit-constrained, here defined as one if a household needed credit but was unable to get and zero otherwise [46], are likely to have lower WTP for TRC since credit is expected to ease liquidity constraints that farmers experience due to imperfect rural markets [44].

Social capital and networks are vital for creating awareness and facilitating the exchange of information, access to inputs, and overcome constraints especially in rural areas where sources of information are inadequate and markets are imperfect [44, 48]. A large proportion (87%) of the sampled households had a household member who participated in a rural institution, which may positively influence the WTP for TRC.

Existing literature shows that farmers' knowledge and perceptions regarding a production constraint as well as characteristics of an innovation that respond to the constraint are likely to influence their adoption decisions [26]. About 22% of the survey respondents could identify at least four major clinical symptoms of AAT. Accurate identification of AAT symptoms suggests correct diagnosis and required management and thus may positively influence the WTP for TRC. Similarly, only a few (8%) of the respondents were aware of the negative effects of trypanocides, which may negatively influence farmers' decision to seek alternative treatment methods such as TRC and thus lower WTP. Positive perception towards the effectiveness of trypanocides may negatively affect the WTP for tsetse collars, as farmers may not be convinced of extra spending on the already working solution. About half of the respondents perceived trypanocides to be effective, while 16% perceived AAT prevalence to be high. Awareness of *icipe* TRC was high (91%), which is expected to influence WTP positively.

## 3.2 Livestock production and constraints

Livestock is a key asset in Kwale County, as demonstrated during qualitative and quantitative surveys, and it is ranked second after crop production in terms of economic importance [35]. Table 2 shows the ownership of major livestock types owned by the surveyed respondents. About 84% of the respondents (533) owned an average of three indigenous cows, while only 15 households had an exotic or a crossbreed cow. About 26% of them reported that they did not own any cows before *icipe* initiated the field experiment on TRC in the area. About 79% of the respondents owned oxen, which are kept mainly for draft power and for slaughter. Fifty-three (53%) of the FDG respondents reported that their main reason for keeping cattle was for draft

**Table 2. Livestock ownership among the surveyed sample in Kwale County.**

| Sample = 632 | N | Mean | Std. Dev | [95% Conf. interval] | |
|---|---|---|---|---|---|
| Indigenous cows | 535 | 3.17 | 3.19 | 2.90 | 3.44 |
| Exotic or cross breed | 15 | 1.80 | 1.90 | 0.75 | 2.85 |
| Oxen | 498 | 2.80 | 1.24 | 2.69 | 2.91 |
| Bulls | 193 | 2.15 | 1.49 | 1.94 | 2.36 |
| Heifers | 252 | 2.29 | 1.84 | 2.06 | 2.52 |
| Calves | 361 | 2.23 | 1.61 | 2.06 | 2.40 |
| Sheep | 101 | 5.26 | 4.23 | 4.42 | 6.10 |
| Goats | 471 | 7.67 | 6.15 | 7.11 | 8.23 |

Source: Household survey

use, closely corroborating with Mbahin et al. [34] and Ohaga et al. [49]. On average, oxen ownership had increased by about 20% after the introduction of the *icipe* TRC.

**3.2.1 Livestock pests and diseases.** Tsetse fly infestation and infection with Trypanosomosis were reported as the most important livestock production constraints in the project area (Table 3). Ticks and tick-borne diseases such as East Coast Fever (ECF) followed closely and were reported by 91% and 36% of the survey respondents, respectively. Worms and small biting flies were also recognized as important pests, reported by a significant number of survey respondents. Animal licking of soil was identified as an important constraint, reported by 68% of the survey respondents. The FDG respondents associated this problem with a deficiency of certain body nutrients. Although soil ingestion by livestock may contribute to the intake of essential minerals [50], it may be detrimental because of increased tooth wear, infections in the digestive tract, or cause indirect effects of harmful chemicals absorbed through the soil particles [51]. Pneumonia was also observed to be an important constraint, reported by 41% of the respondents, being higher than those reported by Ohaga et al. [49] (8%) and suggesting an increased prevalence of the disease in the study area.

**3.2.2 Availability and utilization of livestock management technologies.** The majority of the respondents reported that curative technologies were available (94%), while vaccination

**Table 3. Livestock pests and diseases in Kwale County.**

| Pests | N = 632 Percent (%) |
|---|---|
| Tsetse Flies | 90.03 |
| Ticks | 90.82 |
| Worms | 79.59 |
| Small biting flies | 59.02 |
| **Diseases** | |
| Trypanosomiasis | 85.76 |
| Soil licking | 68.35 |
| Pneumonia | 40.51 |
| East Coast Fever | 36.55 |
| Skin diseases | 27.22 |
| Eye disease | 24.37 |
| Anthrax | 12.03 |
| Brucellosis | 5.06 |

Source: Household survey

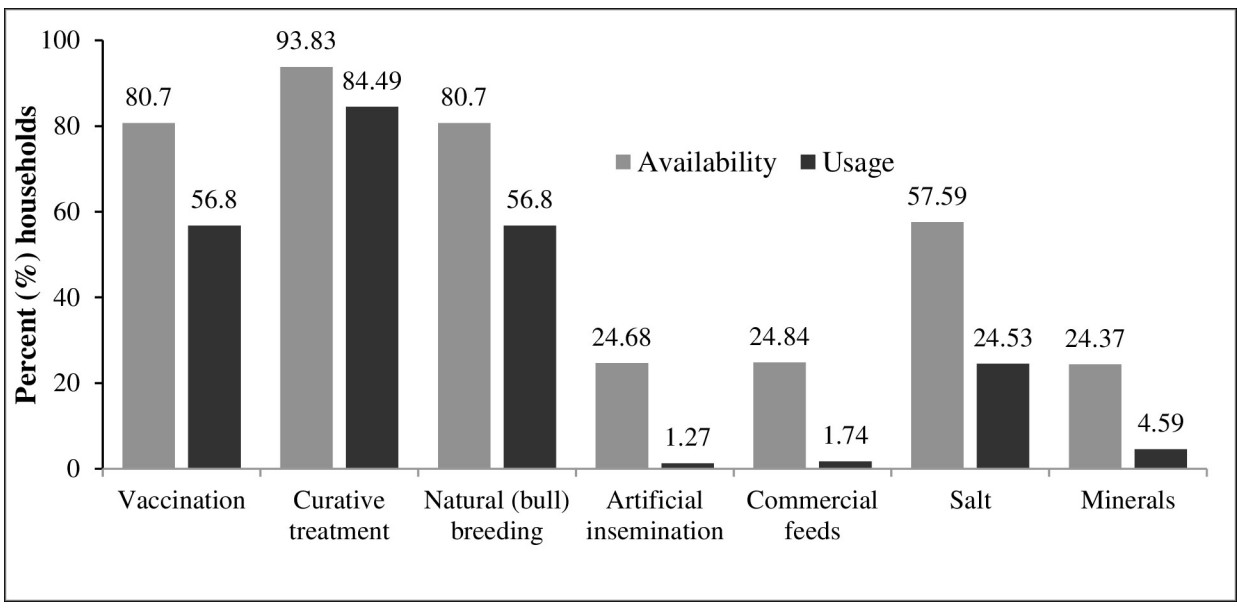

**Fig 2. Availability and use of livestock management technologies in Kwale County.**

and natural breeding technologies were reported by 81% of the respondents (Fig 2). On utilization, about 84% of the households had used curative treatment, while about 57% had used vaccination and natural bull breeding technologies (Fig 2). Other technologies such as artificial insemination and commercial feeds were reported by about 25% of the respondents as available and yet only 2% had utilized them (Fig 2). About 19% of the respondents reported that they had received training on tsetse and AAT management, while 17% and 6% reported receiving training on managing other diseases and livestock breeding, respectively.

### 3.3 Farmer's knowledge, perception, and management of tsetse and Trypanosomosis

**3.3.1 Knowledge of Trypanosomosis clinical symptoms.** About 74% of the respondents listed starring coat, followed by recumbency (45%), emaciation (31%), fever (23%), and low milk production (21%) as the most common symptoms of Trypanosomosis (Table 4). About 22% of the respondents reported more than four clinical symptoms of AAT, and only 9% stated more than five signs, which is significantly lower than Ohaga et al. [49]. This implies that increased awareness of the clinical symptoms of the disease is required in this region. Other identified minor signs included diarrhoea (13%), abortion (11%), and anorexia (9%), some of which have been identified in the previous studies [36, 49].

**3.3.2 Perceived tsetse fly infestation and cattle mortality rate.** Out of the 90% that responded positively, when asked if tsetse attacked their livestock (Table 3), 16% rated the prevalence of AAT as high and associated 8% of cattle mortality with the disease (Table 5). The majority of respondents rated the disease prevalence as medium (35%) and low (37%), while 56% associated a low mortality rate with disease prevalence. This perceived infestation and disease severity (average 29%) were higher than the 16% reported by Machila et al. [52]. Using the number of tsetse catches per trap, Ohaga et al. [49] reported tsetse infestation of 18%, whereas Mbahin et al. [34] and Muraguri et al. [53] reported 34% and 29%, respectively. This finding suggests that despite the notable effort to control tsetse and manage AAT in Kwale, the disease remains a significant challenge in livestock production in the area.

**Table 4. Clinical signs trypanosomiasis reported in Kwale district (n = 632).**

| AAT Clinical sign (N = 632) | Farmers reporting the symptoms (%) |
|---|---|
| Starring coat | 82.4 |
| Recumbency | 45.4 |
| Emaciation | 31.0 |
| Fever | 22.6 |
| Low milk production | 21.2 |
| Diarrhoea | 13.4 |
| Abortion | 10.9 |
| Death | 8.9 |
| Anorexia | 6.3 |
| Coughing & running nose | 5.2 |
| Physical weakness | 3.3 |
| Lacrimation of eyes | 2.8 |
| Salivation | 2.7 |
| Swelling of lymph nodes | 2.7 |
| Bloody spots on the skin (chancre) | 2.4 |
| Running around | 1.3 |
| Eating soil | 0.9 |
| Constipation | 0.6 |

Source: Household survey

**3.3.3 Tsetse management practices.** Avoiding grazing in tsetse-infested areas, especially near the Shimba hills game reserve, was the most popular method of preventing tsetse flies bites as reported by 79% of survey respondents (Table 6). This closely collaborates with Ohaga et al. [49] results. Following closely, was the use of trypanocidal drugs reported by 76% of respondents, which was consistent with previous studies conducted in the study site [34, 49, 52, 53]. Other commonly used cultural methods included avoiding grazing animals during the high-risk periods (early morning and late evening) and clearing and burning bushes surrounding the cattle sheds. These strategic shifts in the grazing period as an intervention against tsetse were also reported by Seyoum et al. [36].

**3.3.4 Knowledge and use of *icipe* tsetse repellent collars technology for the management of AAT.** Having conducted a field trial of the tsetse repellent technology in Kwale County [16], many survey respondents (89%) were aware of the *icipe*'s technology and about 42%

**Table 5. Perceived tsetse infestation and severity of trypanosomiasis.**

| | Survey respondents (%) |
|---|---|
| *Prevalence* | |
| High | 16.14 |
| Medium | 34.81 |
| Low | 37.18 |
| *Mortality* | |
| High | 8.39 |
| Medium | 24.21 |
| Low | 55.54 |

Source: Household survey

**Table 6. Tsetse fly and trypanosomiasis management strategies in Kwale County.**

|  | Survey respondents (%) |
|---|---|
| Avoiding tsetse infested areas | 79.09 |
| Injecting animals with trypanocides (drugs) | 68.04 |
| Delaying time of taking cattle to graze | 26.27 |
| Spraying/dipping sick animals | 21.68 |
| Pour-ons technology | 11.22 |
| Clearing bushes | 10.13 |
| Grazing near animals with *icipe* collars | 8.70 |
| Grazing far from other animals | 5.70 |
| Netted zero-grazing units | 4.05 |
| Smearing animal skin with ash or ointments | 3.80 |

Source: Household surveyss

reported ever using the technology. A smaller percentage (30%) has continued to use the experimental TRC in an effort by the project to determine their cost-effectiveness. Other community-level tsetse control strategies experimented alongside the *icipe* TRC include the baited NGU traps [54] and target screens [16]. About 88% and 67% of the survey respondents were aware of these strategies, respectively. Furthermore, over 20% of those who had to use the additional technologies believed they effectively reduced AAT transmission.

## 3.4 Willingness to pay for an icipe repellent collar for management for tsetse flies

Sixty-three (63%) percent of the interviewed farmers were WTP for the new *icipe* TRC at a price similar to their current cost of treating an animal for tsetse and Trypanosomosis (without the repellent collar), estimated at KES 2,673 per animal per year. The rest rejected the offer. They were then offered a discount, ranging from 15% to 60% of their current cost of treating one animal per year, randomly picked by the data collection program (Cspro) and presented to the farmer. Once offered a discount, additional farmers were willing to buy the TRC (dark shaded bars on the left side of Fig 3). For instance, out of those offered a 30% discount, 39% of them would eventually accept the offer, suggesting an increase of 7% (dark shaded part of the third bar of Fig 3) to those who accepted the initial bid. Those who accepted the first bid (63%) were then asked whether they would be WTP for the collar at a higher price, ranging from 15% to 60% of the initial bid. About 14% were willing to accept a 30% premium (seventh bar of Fig 3). When asked when they would buy the collars, of those farmers that expressed willingness to pay, 83% would buy immediately while 14% would buy after one year. Asked if they would cover their entire herd, at the initial purchase, farmers reported that they would buy collars for an average of three heads of cattle, while the rest would be covered after one year.

## 3.5 Mean price that farmers are WTP for tsetse repellent collar and factors influencing WTP in Kwale County

The mean WTP was estimated using the logistic model (Eq (1)). We first estimate the unconditional model where we assume that socioeconomic characteristics do not affect the WTP, then we include the explanatory variables that may also affect the WTP for the TRC. The results are given in Tables 7 and 8. Note that the sample of farmers used in the regression (Tables 7 & 8) is less than the earlier sample utilized in the descriptive statistics. Four observations were

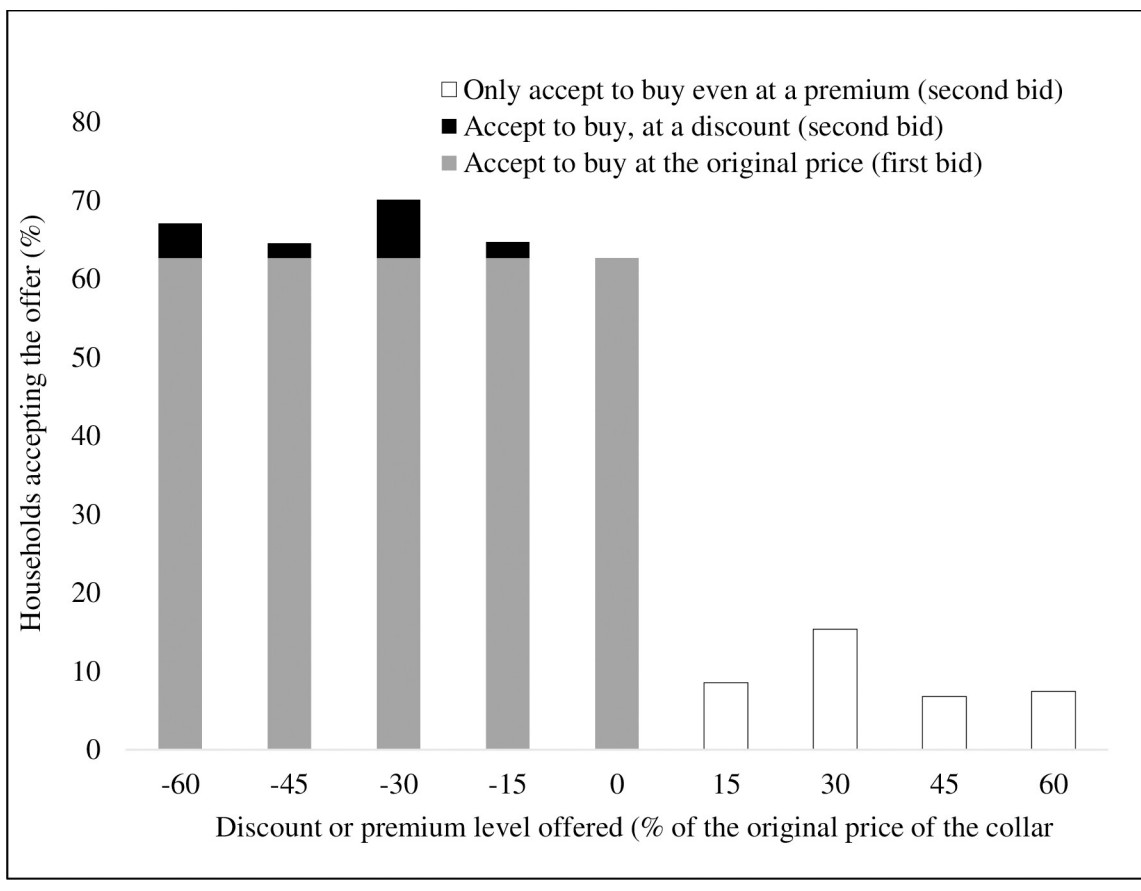

**Fig 3. Willingness to pay for the *icipe* tsetse fly repellent collar in Kwale County.**

dropped from the analysis due to missing responses on the WTP for the tsetse repellent collar. As shown in Table 7 the WTP was for the *icipe* collar was KES 3,352, about 25% over and above the annual cost of treating one mature animal for tsetse flies and AAT-related complications.

The conditional analysis (Eq (3)) includes other explanatory variables that are likely to affect the WTP for TRC. Controlling for the explanatory variables reduced the mean WTP slightly KES 3,313 (Table 8) but remained significantly higher (23%) than the annual cost of

**Table 7. Logistic parameter estimates for WTP for tsetse repellent collar (without socio-economic characteristics) in Kwale County.**

| Variable | Coefficient | Standard Error |
| --- | --- | --- |
| Sigma/Bid | 3458.55*** | 155.41 |
| Mean WTP | 3351.66*** | 160.56 |
| Number of observations | 628 | |
| Log-likelihood | -1003.458 | |

*p < 0.1

**p < 0.05

***p < 0.01. *Note*: Exchange rate during the period of the survey was KES 100 for 1US$.

**Table 8. Factors influencing the WTP for *icipe's* tsetse repellent collar in Kwale county.**

|  | Coefficient | Std. Err. |
|---|---|---|
| **Mean WTP** | 3313.36*** | 151.63 |
| *Household characteristics* | | |
| Gender | 839.45* | 436.16 |
| Age | -13.05 | 12.40 |
| Education level | 70.67* | 43.83 |
| Household size | -168.91 | 158.44 |
| *Household resources* | | |
| Livestock (TLU) | 11.62 | 37.98 |
| Farm Size (hectares) | 62.45 | 51.98 |
| Main occupation (1 = farming, 0 = otherwise) [a] | 48.15 | 417.60 |
| Off-farm income | 250.27 | 357.49 |
| *Access to market and institutional services* | | |
| Livestock training | 186.67 | 398.69 |
| Extension proximity | 0.78 | 1.83 |
| Credit constrained | 499.72 | 312.44 |
| Market distance | 1.05 | 1.29 |
| *Social capital and networks* | | |
| Rural institutions | -503.85 | 451.86 |
| *Knowledge and perceptions* | | |
| AAT clinical symptoms[b] | 485.74 | 387.49 |
| Negative chemical effects | -55.64 | 578.62 |
| Trypanocides effectiveness | 26.58 | 318.21 |
| AAT prevalence | 2322.18*** | 424.42 |
| Aware of tsetse collar | 385.60 | 558.64 |
| Constant | 1501.90 | 1417.48 |
| Sigma (σ) | 3237.41*** | 145.63 |
| Number of observations | 628 | |
| Wald chi2 (18) | 48.58 | |
| Log-likelihood | -979.59 | |
| Prob >chi2 | 0.0001 | |

*p < 0.1

**p < 0.05

***p < 0.01. *Note*: Exchange rate during the period of the survey was KES 100 for 1US$.

[a]Farming is compared to the other three occupations collapsed together due to few observations.

[b]A dummy variable equal to 1 if a farmer identified four clinical symptoms correctly.

treating an animal for AAT. Furthermore, the mean WTP was over 100% higher than the annual cost of the proposed *icipe* TRC (KES 1,475) (S1 Text).

Table 8 also shows the variables that we controlled for in the WTP model. Concerning household characteristics, the regression results show that gender and education of the household head significantly and positively influence WTP. Relative to a female-headed household, a household headed by male farmers was willing to pay more for TRC by about KES 840. This result suggests that male heads are more likely to adopt TRC compared to female heads. The finding was expected as female heads are often associated with limited access to production resources to facilitate the adoption of new agricultural innovations [39]. Similarly, household heads with more years of schooling are more likely to adopt TRC than those with less

education. An additional year of education increases the WTP for TRC by about KES 71. Education enhances the processing and interpretation of new information to address production constraints and increase farm returns [40]. None of the household resource proxies was significant, neither were the variables proxied for access to market and institutional services and social capital and networks. Among the knowledge and perceptions variables, the coefficient for the prevalence of AAT is positive and significant. High perceived AAT prevalence increases WTP for TRC by about KES 2,322. This is plausible, as farmers would be willing to adopt any solution to reduce tsetse infestation and the prevalence of AAT to maximize returns from their animals.

Thirty-seven (37%) of the surveyed households who responded negatively to the WTP question (Fig 3), were asked to state the reasons for their responses. Over half (52%) of them indicated that the technology was expensive for them. This amplifies resource constraints among smallholder farmers and limits them from acquiring even the readily available tools and solutions for addressing their farming challenges. Other reported challenges included lack of technical support on the application of the technology, limited herd sizes, perceived reduction of tsetse infestation, and a generally negative attitude towards the effectiveness of the technology. The latter challenges can be addressed through wider dissemination and training on the technology application.

## 4 Discussion and conclusion

### 4.1 Tsetse and Trypanosomosis awareness, perception, and management practices

The economic importance of tsetse and AAT (90% and 86%, respectively) in the study area is supported by earlier studies [49, 52]. This is attributable to Shimba Hills National Reserve being a major harbour of the disease vectors sustained by the alternative wild hosts. Tsetse flies and AAT have also been reported as an important constraint to livestock production in other African Countries ([36, 55]. Contradicting with earlier studies, e.g. Ohaga et al. [49], there is a knowledge gap in the identification of clinical signs of AAT since only a small proportion (31%) of the respondents identified more than four symptoms. Inaccurate diagnosis of the disease may suggest wrong treatment and/or improper timing of the treatment, as also noted by Machila et al. [52], Ohaga et al. [49], and Muraguri et al. [53]. The perceived prevalence of AAT (29%) in the area corroborates with previous studies [34, 49, 52, 53]. However, while our finding compares well with those of Mbahin et al. [34] and Muraguri et al. [53], the others report lower perceived disease prevalences. The earlier studies might have underestimated the burden of AAT due to the limited size of the survey respondents. Nevertheless, in our study, the majority of the respondents rated AAT prevalence low (37%), with only 16% ranking it as high. The low prevalence rate was associated with *icipe*'s effort in introducing the household and community level tsetse vector control (TRC and baited targets) measures. The FDG respondents noted that, on average the AAT prevalence had reduced from 85% animals falling sick with AAT in a household, to about 20% after introduction of these measures about seven years ago. Similarly, a significant number of our survey respondents rated livestock mortality due to AAT as low (56%), which was consistent with our qualitative findings that reported about 5% mortality rate. The study therefore supports upscaling efforts of the tsetse control technology for sustainable management of the AAT vector and the disease.

Cultural practices, such as avoiding grazing animals in areas perceived to be tsetse-infested such as grasslands next to the SHNR, remain important for reducing AAT transmission. This finding is different from that of Ohaga et al. [49] who reported that farmers did not associate grazing near SHNR with trypanosomosis incidence but rather with the timing of the grazing

periods. Seyoum et al. [36] observed that riverside, forest, bushy grassland, and grazing area were the riskiest places for tsetse fly and AAT exposure. This is further supported by the biology and ecology of the flies in literature [56]. Our results, like those of Ohaga et al. [49], indicate that the timing of grazing periods is still considered to be an effective tsetse management mechanism thus (27%) corroborating with Catley & Leyland [19] and Tesfaye et al. [57]. These cultural methods could be promoted alongside the tsetse control technologies.

The extensive use of trypanocidal drugs (68%) in the area is supported by earlier research findings [34, 49, 52, 53]. However, while earlier studies reported inappropriate use of trypanocidal drugs that increasingly result in drug resistance (e.g. Machila et al. [52]), our qualitative findings reported improved knowledge in treating animals using the trypanocidal drugs, including diagnosis, determining the weight of the animal that corresponds to a certain quantity of the drug, as well as the timing for the drug application. This has been achieved through training offered to the farmers in the area by *icipe* and partners over the years of the experimental trial of the AAT vector targeted management strategies.

A notable implication of the above results pertains to the knowledge gap in the identification of the AAT disease. Although the disease diagnosis has improved over time, only a small proportion of the sampled farmers could correctly identify at least four key symptoms of the disease. These results, therefore, suggest the need for further training and awareness creation for correct diagnosis of the disease and thus accurate treatment, including the use of TRC.

## 4.2 Potential uptake of tsetse repellent collar technology

We found a significant number of respondents (63%) were willing to pay for *icipe* TRC at a cost equivalent to treating an animal for AAT using trypanocidal drugs. Furthermore, the overall mean WTP, even after controlling for household exogenous variables, was significantly higher (23–25%) than the annual cost of treating animals for AAT without the collar, and about 124% higher than the approximate annual cost of the collar. The positive WTP of the tsetse management innovation supports earlier finding such as Saini et al. [16] and Walubengo et al. [58] in Kenya, and Seyoum et al. [36] in Ethiopia. Our study indicates that commercialized TRC will elicit a high and suitable market among livestock farmers.

The assessment of socioeconomic characteristics represented the differential influences of several factors on farmer's WTP. The gender and education of the household head for instance, positively correlated with WTP. The results are in agreement with previous studies on agricultural technology adoption, that female heads and less educated farmers are less likely to adopt new technologies, due to limited access and use of productive resources [39], and less able to acquire, absorb, interpret and use new information to implement new technology [59], respectively.

The knowledge and perception of a production constraint and the characteristics of new technology are likely to influence the adoption of the technology. Among the knowledge and perception proxies of this study, we find a significant correlation between perceptions of AAT prevalence and WTP for the TRC. This agrees with previous reports, for instance, Campbell et al. [60] who found knowledge score on new castle disease and vaccines to increase willingness to pay for vaccination services in Tanzania.

An important implication of the above findings is that there is potential demand for the TRC technology. A sizable number of the farmers were willing to pay for the TRC at the same cost of trypanocidal drugs. This amount is significantly over and above the approximate annual cost of the novel repellent collar. Enhancement of this potential demand and access to the technology among farmers will therefore involve increased sensitization of commercial dealers of veterinary products. These efforts will require targeting women farmers and the

relatively little educated to influence their WTP for the technology. Women can be efficiently reached through women's group targeted training and other forums that predominantly involve women farmers.

While we find valuable insights into the potential adoption of the novel repellent collar for the control of tsetse flies and AAT transmission, our study focussed on livestock farmers, ignoring input suppliers in the value chain. Elicitation of the willingness to stock the TRC among the commercial traders would answer critical policy questions regarding the adoption of the technology. Second, we use cross-sectional data, and therefore unable to model TRC demand dynamics. These shortcomings should be the subject of future research that could strengthen the evidence of this paper.

## Supporting information

**S1 Text. Willingness to pay for a tsetse repellent collar for control of tsetse flies and Trypanosomosis transmission.** A narrative of the procedure followed to elicit the respondent's willingness to pay for the novel tsetse repellent collar.
(DOCX)

**S1 Fig. icipe tsetse repellent collars. (a)** New canvas tsetse repellent collar that is about to be commercialized. (b) Earlier developed tsetse repellent collar (Saini et al., 2017) [16]
(TIF)

**S1 File. Data processing strategy: Stata codes.**
(PDF)

**S2 File. Data set: Stata file.**
(DTA)

## Acknowledgments

We are highly indebted to Dr. Menale Kassie for his advice and support, the Animal health team of icipe for field support, the enumerators and survey supervisor for their effort in data collection, and the cattle keepers in Kwale County for their time and invaluable information for this study.

## Author Contributions

**Conceptualization:** Beatrice W. Muriithi, Gracious M. Diiro, Michael M. Kidoido, Michael Nyangánga Okal, Daniel K. Masiga.

**Data curation:** Beatrice W. Muriithi, Nancy G. Gathogo, Gracious M. Diiro.

**Formal analysis:** Beatrice W. Muriithi, Nancy G. Gathogo.

**Funding acquisition:** Beatrice W. Muriithi, Gracious M. Diiro, Michael Nyangánga Okal, Daniel K. Masiga.

**Investigation:** Beatrice W. Muriithi, Nancy G. Gathogo, Gracious M. Diiro, Daniel K. Masiga.

**Methodology:** Beatrice W. Muriithi, Nancy G. Gathogo, Gracious M. Diiro, Michael M. Kidoido.

**Project administration:** Beatrice W. Muriithi, Nancy G. Gathogo, Gracious M. Diiro, Michael M. Kidoido, Michael Nyangánga Okal, Daniel K. Masiga.

**Resources:** Beatrice W. Muriithi, Michael Nyangánga Okal, Daniel K. Masiga.

**Software:** Beatrice W. Muriithi, Nancy G. Gathogo.

**Supervision:** Beatrice W. Muriithi, Gracious M. Diiro, Michael M. Kidoido.

**Validation:** Beatrice W. Muriithi.

**Visualization:** Beatrice W. Muriithi, Daniel K. Masiga.

**Writing – original draft:** Beatrice W. Muriithi, Nancy G. Gathogo, Gracious M. Diiro, Michael M. Kidoido, Michael Nyangánga Okal, Daniel K. Masiga.

**Writing – review & editing:** Beatrice W. Muriithi, Nancy G. Gathogo, Gracious M. Diiro, Michael M. Kidoido, Michael Nyangánga Okal, Daniel K. Masiga.

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
