## [Decision Letter · Decision Letter 0]

27 Apr 2021

Dear DR Muriithi,

Thank you very much for submitting your manuscript "Knowledge, Perceptions, and Willingness to Pay for a Tsetse Repellent Collar among Smallholder Livestock Farmers in Kwale County, Kenya" for consideration at PLOS Neglected Tropical Diseases. As with all papers reviewed by the journal, your manuscript was reviewed by members of the editorial board and by several independent reviewers. In light of the reviews (below this email), we would like to invite the resubmission of a significantly-revised version that takes into account the reviewers' comments. 

We cannot make any decision about publication until we have seen the revised manuscript and your response to the reviewers' comments. Your revised manuscript is also likely to be sent to reviewers for further evaluation.

Sincerely,

Adly M.M. Abd-Alla, Prof asso.

Associate Editor

Fabiano Oliveira

Deputy Editor

Reviewer's Responses to Questions

**Key Review Criteria Required for Acceptance?**

**Methods**

-Are the objectives of the study clearly articulated with a clear testable hypothesis stated?

-Is the study design appropriate to address the stated objectives?

-Is the population clearly described and appropriate for the hypothesis being tested?

-Is the sample size sufficient to ensure adequate power to address the hypothesis being tested?

-Were correct statistical analysis used to support conclusions?

-Are there concerns about ethical or regulatory requirements being met?

Reviewer #1: See attached report

Reviewer #2: The objectives and hypothesis are not outlined clearly. 

The study design is well explained and suitable, the selection of the respondents followed a good and robust procedure - a probability sampling was used at the last stage where a the farmers were drawn from a sampling frame. The sample size is adequate and the statistical techniques are suitable. However there was mention of qualitative data collection and triangulation in the methods, but this was not indicated in the summary or abstract. If a mixed method was used it should be appear prominently.

There are no concerns about ethical issues.

Reviewer #3: -Are the objectives of the study clearly articulated with a clear testable hypothesis stated? Yes

 -Is the study design appropriate to address the stated objectives? Yes

-Is the population clearly described and appropriate for the hypothesis being tested? Yes

-Is the sample size sufficient to ensure adequate power to address the hypothesis being tested? Yes

-Were correct statistical analysis used to support conclusions? Yes

-Are there concerns about ethical or regulatory requirements being met? No

**Results**

-Does the analysis presented match the analysis plan?

-Are the results clearly and completely presented?

-Are the figures (Tables, Images) of sufficient quality for clarity?

Reviewer #1: See attached report

Reviewer #2: The results are well presented. However, line 491- 492 refers to 56% of respondents as majority, but this figure is about half or slightly more than half - it might be misleading therefore it should be rephrased.

Reviewer #3: -Does the analysis presented match the analysis plan? Yes

-Are the results clearly and completely presented? Yes, but it would be very convenient to add the confidence interval estimate to draw conclusions about the population characteristics from the sample results.

-Are the figures (Tables, Images) of sufficient quality for clarity? Yes

**Conclusions**

-Are the conclusions supported by the data presented?

-Are the limitations of analysis clearly described?

-Do the authors discuss how these data can be helpful to advance our understanding of the topic under study?

-Is public health relevance addressed?

Reviewer #1: See attached report

Reviewer #2: The public health relevance should be highlighted as well as the limitations of the study. The authors have made a good case of how the study can help address the constraints of testes fly pest in livestock development. Icipe should be written in full at first time usage.

Reviewer #3: -Are the conclusions supported by the data presented? Yes

-Are the limitations of analysis clearly described? No

-Do the authors discuss how these data can be helpful to advance our understanding of the topic under study? Yes

-Is public health relevance addressed? Yes

**Editorial and Data Presentation Modifications?**

Reviewer #1: See attached report

Reviewer #2: I recommend the manuscript can proceed with minor revisions. A few grammatical errors appear - line 16 and 20. These should be addressed.

Reviewer #3: (No Response)

**Summary and General Comments**

Reviewer #1: See attached report

Reviewer #2: The authors have made a relevant and good case of how the study can help address the constraints of testes fly pest in livestock development. The use of a sampling fame to select respondents added rigor and strength to the study. However the study objectives should be clearly articulated. Minor observations should be addressed.

Reviewer #3: REVIEW 

Apr26, 2021.

This interesting investigation of Muriithi and Cols. is relevant for the environmental and public health, based in the previous study from icipe and by applying the contingent valuation (CV) method to elicit farmer's willingness to pay (WTP) for the novel Tsetse repellent collar (TRC) using data collected from a representative sample of cattle keepers in Kwale county, Kenya. Wide-scale commercialization and adoption of the repellent collar technology for lowing AAT transmission will depend on farmers' pre-conceived perceptions, knowledge and notably their willingness to pay, for the technology.

This technological innovation preventing disease transmission: a new paradigm for vector control. Evidence that deploying water buck repellents converts cattle into non-hosts for tsetse flies-'cows in waterbuck clothing, Saini and Cols, 2017.

Important problems:

There is not commented about limitation of study.

Refresh references. I suggest it could be important.

Observation:

1) Add the keyword: novel Tsetse repellent collar.

2)I consider important to add the effectiveness, safety

 and mechanism of action of Tsetse repellent collar in the introduction.

3) It is suggested to add confidence interval in Tables 1 and 2.

4) Page 8 File 181: Which version of STATA was used?

5) I like to know about study participants: They know how to read?

If not: How answer the survey? There were excluded?

 Reference:

1. Jemberu WT, Molla W, Dagnew T, Rushton J, Hogeveen H (2020) Farmers’ willingness to pay for foot and mouth disease vaccine in different cattle production systems in Amhara region of Ethiopia. PLoS ONE 15(10): e0239829. https://doi.org/10.1371/journal. pone.0239829 

2. Adungo, F., Mokaya, T., Makwaga, O., & Mwau, M. (2020). Tsetse distribution, trypanosome infection rates, and small-holder livestock producers' capacity enhancement for sustainable tsetse and trypanosomiasis control in Busia, Kenya. Tropical medicine and health, 48, 62. https://doi.org/10.1186/s41182-020-00249-0

3. Patterns of tsetse abundance and trypanosome infection rates among habitats of surveyed villages in Maasai steppe of northern Tanzania. Ngonyoka A, Gwakisa PS, Estes AB, Salekwa LP, Nnko HJ, Hudson PJ, Cattadori IM.Infect Dis Poverty. 2017 Sep 4;6(1):126. doi: 10.1186/s40249-017-0340-0.

4. Olaide OY, Tchouassi DP, Yusuf AA, Pirk CWW, Masiga DK, Saini RK, Torto B. Zebra skin odor repels the savannah tsetse fly, Glossina pallidipes (Diptera: Glossinidae). PLoS Negl Trop Dis. 2019 Jun 10;13(6): e0007460. doi: 10.1371/journal.pntd.0007460. PMID: 31181060; PMCID: PMC6586361.

5. Masiga DK, Igweta L, Saini R, Ochieng'-Odero JP, Borgemeister C. Building endogenous capacity for the management of neglected tropical diseases in Africa: the pioneering role of ICIPE. PLoS Negl Trop Dis. 2014 May 15;8(5): e2687. doi: 10.1371/journal.pntd.0002687. PMID: 24830708; PMCID: PMC4022455.

PLOS authors have the option to publish the peer review history of their article (what does this mean?). If published, this will include your full peer review and any attached files.

Reviewer #1: Yes: Dr. Daniel Kyalo Willy

Reviewer #2: Yes: Fatima Abdulaziz

Reviewer #3: No
---

## [Decision Letter · Decision Letter 1]

19 Jul 2021

Dear DR Muriithi,

We are pleased to inform you that your manuscript 'Farmer Perceptions and Willingness to Pay for Novel Livestock Pest control Technologies: A case of Tsetse repellent Collar in Kwale County in Kenya' has been provisionally accepted for publication in PLOS Neglected Tropical Diseases.

Best regards,

Adly M.M. Abd-Alla, Prof asso.

Associate Editor

Fabiano Oliveira

Deputy Editor

Reviewer's Responses to Questions

**Key Review Criteria Required for Acceptance?**

**Methods**

-Are the objectives of the study clearly articulated with a clear testable hypothesis stated?

-Is the study design appropriate to address the stated objectives?

-Is the population clearly described and appropriate for the hypothesis being tested?

-Is the sample size sufficient to ensure adequate power to address the hypothesis being tested?

-Were correct statistical analysis used to support conclusions?

-Are there concerns about ethical or regulatory requirements being met?

Reviewer #1: I have reviewed the revised version of the paper and feel that the comments have been addressed appropriately.

Reviewer #2: The study has clear set objectives and the methods appear sound. The revisions made addresses all issues.

Reviewer #3: No comments.

**Results**

-Does the analysis presented match the analysis plan?

-Are the results clearly and completely presented?

-Are the figures (Tables, Images) of sufficient quality for clarity?

Reviewer #1: The analysis matched the plan appropriately. The comments raised earlier on the section have been addressed.

Reviewer #2: The results are well articulated.

Reviewer #3: No comments.

**Conclusions**

-Are the conclusions supported by the data presented?

-Are the limitations of analysis clearly described?

-Do the authors discuss how these data can be helpful to advance our understanding of the topic under study?

-Is public health relevance addressed?

Reviewer #1: The conclusions read better now and have reflected the results.

Reviewer #2: The conclusion is well drawn and the authors have illuminated limitations as well as the public health relevance.

Reviewer #3: Study limits

Consider that, due to the universe of the study sample, it may present a selection bias and contribute to overestimate the perception and willingness to pay for this new technology (novel tsetse fly repellent collar), since the participants were chosen from the same population from the previous study where it was tested, so those who noticed a benefit may be more interested in this alternative. I agree with the authors that future studies will help in external validity where the choice of participants is random and without a history of having tried its use, ideally including both sexes, with and without education, which is a social determinant of interest. Another limit may be territorial distances, due to communication in places that are difficult to access, possibly related to a higher poverty rate. As a perspective, studies on survey and follow-up have recently been published through the use of telephone interviews, since in the African continent the use of cell phones has increased notably in recent years, so it is suggested to consider their application for future research .

**Editorial and Data Presentation Modifications?**

Reviewer #1: No major revisions are needed. The editorial issues noted in the initial draft have been addressed appropriately. Just obe comment: Avoid joint interpretation of variables, unless joint analysis was done. In the abstract the Authors state: "Male educated household heads are likely to pay more for the TRC." This is not appropriate since we don't see any estimation of the joint effect of Gender and Education level in the analysis.

Reviewer #2: Accept

Reviewer #3: The current version of the article is clearer and more understandable with noticeable improvement in its content. The observations made to the first version have been answered, I only add a suggestion for the authors regarding the limits of the study. Accepted.

**Summary and General Comments**

Reviewer #1: Proceed to accept the paper.

Reviewer #2: The reviewed manuscript is well executed from the background and methods. Clear results and conclusions have been articulated.

Reviewer #3: No comments.

PLOS authors have the option to publish the peer review history of their article (what does this mean?). If published, this will include your full peer review and any attached files.

Reviewer #1: **Yes: **Dr. Daniel Kyalo Willy

Reviewer #2: **Yes: **Fatima Abdulaziz Sule

Reviewer #3: **Yes: **Rocio A Castillo-Cruz

---

## [Editor Report · Acceptance letter]

10 Aug 2021

Dear DR Muriithi,

We are delighted to inform you that your manuscript, "Farmer Perceptions and Willingness to Pay for Novel Livestock Pest control Technologies: A case of Tsetse repellent Collar in Kwale County in Kenya," has been formally accepted for publication in PLOS Neglected Tropical Diseases.

Best regards,

Shaden Kamhawi

co-Editor-in-Chief

Paul Brindley

co-Editor-in-Chief
